# Retrospective Long-Term Evaluation of Miltefosine-Allopurinol Treatment in Canine Leishmaniosis

**DOI:** 10.3390/pathogens12070864

**Published:** 2023-06-22

**Authors:** Manuela Gizzarelli, Valentina Foglia Manzillo, Antonio Inglese, Serena Montagnaro, Gaetano Oliva

**Affiliations:** 1Department of Veterinary Medicine and Animal Production, University of Naples Federico II, 80137 Naples, Italy; manuela.gizzarelli@unina.it (M.G.); semontag@unina.it (S.M.); gaeoliva@unina.it (G.O.); 2Ambulatorio Veterinario Dr. Antonio Inglese, 74023 Grottaglie, Italy; inglese.antonio.vet@gmail.com

**Keywords:** miltefosine, allopurinol, canine leishmaniosis, treatment

## Abstract

Miltefosine-Allopurinol (MIL-AL) combination is reported to be one of the most effective treatments for canine leishmaniosis, thanks to its oral administration and MIL-documented low impact on renal function. However, MIL-AL is considered a second-choice treatment when compared to meglumine-antimoniate—allopurinol combination, mainly due to the risk of earlier relapses. The aim of this study was to evaluate the efficacy of the MIL-AL protocol during a long-term follow-up with an average duration of nine years. Dogs were living in Southern Italy (Puglia, Italy) in an area considered endemic for Canine leishmaniosis (CanL). Inclusion criteria were clinical and/or clinicopathological signs consistent with CanL; positive result to *Leishmania* quantitative ELISA; and negativity to the most frequent canine vector-borne infections. All dogs received 2 mg/kg MIL for 28 days, and 10 mg/kg AL, BID, for a period varying between 2 and 12 months. Ancillary treatments were allowed according to the clinical condition of the dog. A total clinical score and a total clinicopathological score were calculated at each time point by attributing one point to each sign or alteration present and then by adding all points. Improvement after each treatment was defined by the reduction of at least 50% of the total score. A survival analysis (Kaplan–Meier curve) was performed for quantifying the probability of the events occurring during the study follow-up. The following events were considered: decreased and negative ELISA results; improvement/recovery of the clinical and clinicopathological alterations; and relapse of leishmaniasis. One hundred seventy-three dogs (75f and 98m) were retrospectively included in the study by examining their clinical records since the first diagnosis of CanL. One hundred forty-three (83%) dogs were under five years of age. The mean duration of the follow-up period was 5.4 (±1.1) years with a minimum of 3.2 years and a maximum of 9 years. All dogs received a first treatment of MIL-AL at inclusion; then, during the follow-up course, 30 dogs required a second treatment, 2 dogs required a third treatment and 1 dog required a fourth and a fifth treatment. The mean time interval between the first and the second treatment was 27.2 (±18.3) months. After the first treatment, all dogs had decreased ELISA levels, in an average interval of 2.6 (±1.6) months. One hundred seventy dogs (98%) experienced a clinical improvement (mean time 3.0 ± 4.9 months); 152 (88%) dogs were considered clinically recovered after a mean time of 16.7 ± 13.5 months. A similar trend was observed for clinicopathological alterations; interestingly, proteinuria decreased in most dogs (*p* < 0.0001—Chi-square for trends). Thirty dogs experienced relapses, the earliest after 4.8 months. The mean time without relapse was 90.4 (±2.5) months. In relapsed dogs, the mean time for clinical improvement after the second treatment was 8.6 (±12.6) months, whereas it was 11.0 (±15.4) months for clinicopathological alterations. Five dogs had limited gastrointestinal side effects associated with MIL treatment. The present study confirms that the MIL-AL protocol can be considered one of the most effective treatments for CanL therapy, mainly for its capacity to provide a long-time clinical improvement in a large majority of treated dogs. As reported in the literature, the clinical stabilization of dogs does not occur immediately after treatment, probably due to the particular pharmacokinetic properties of MIL. The efficacy of MIL-AL decreases in dogs that need more than one treatment, suggesting the necessity to alternate anti-*Leishmania* drugs for the treatment of relapses. Side effects were transient and slight, even in dogs that required several treatments.

## 1. Introduction

Canine leishmaniosis (CanL) is a zoonotic disease caused by the protozoan *Leishmania infantum* (Kinetoplastida: Trypanosomatidae) widely spread all over the world and affecting millions of dogs [1,2,3]. The infection can evolve with different outcomes, from a subclinical phase to different clinical stages [4] characterized by high variability of clinical signs and clinical-pathologic alterations [5]. The polymorphism of CanL represents a constant challenge for practitioners, its diagnosis needs an integrated approach based on anamnesis, clinic, and laboratory techniques [6], while the therapy requires patient-tailored choices following international guidelines [4].

Miltefosine is the most discussed drug against CanL, especially regarding long-term effectiveness. This compound was originally studied and classified as an anti-tumor drug [7] and only later was its effectiveness in counteracting CanL discovered. Its action is expressed mainly by (i) inhibiting the biosynthesis of the glycosyl phosphatidyl inositol (GPI) receptor, the key molecule for *Leishmania* intracellular survival; (ii) blocking the synthesis of *Leishmania*-specific phospholipase and protein kinase C; and (iii) causing apoptosis of the parasite [8,9]. Once administered orally, miltefosine is gastrointestinally absorbed and widely distributed in body organs; then, it is metabolized in the liver into choline and choline-containing metabolites and is slowly excreted in the feces [10,11]. Several studies have shown how the miltefosine’s pharmacokinetics determines a low-renal impact, both in terms of microscopic lesions detectable at the tissue level and in serum and urinary biomarkers, making this drug a first-choice treatment for dogs suffering from CanL with kidney damage [11,12,13,14].

Despite its efficacy, the use of miltefosine in monotherapy is not recommended due to a higher risk of therapeutic failure [15] or recurrences after the interruption of treatment [16,17]. Instead, the combined therapy of miltefosine and leishmaniostatic allopurinol is a protocol of documented efficacy [18,19] and is considered among those the first choice for CanL therapy [4,20]. Nevertheless, the Miltefosine-Allopurinol (MIL-AL) protocol is sometimes considered a second-choice treatment when compared to meglumine-antimoniate—allopurinol combination, mainly for the risk of earlier relapses [21,22] or the development of parasite drug resistance [23]. The aim of this study was to evaluate the efficacy of the MIL-AL protocol during a long-term follow-up with an average duration of nine years.

## 2. Materials and Methods

### 2.1. Sample Selection

The retrospective study was performed by examining the clinical forms derived from client-owned dogs referred to a private veterinary clinic located in a region of South Italy (Grottaglie, Puglia Region; Latitude = 40°32′27.8″ N, Longitude =17°26′20.8″ E), which is known as endemic for CanL [24,25].

All *Leishmania* symptomatic dogs with a confirmed serological diagnosis of CanL were selected, treated with the MIL-AL therapy, and clinically monitored during a period of 9 years. The following criteria were considered: dog signalment, date of first diagnosis of leishmaniosis, *Leishmania* serological result, clinical signs, clinicopathological alterations, treatment, and the presence/absence of concomitant diseases.

### 2.2. Ethical Approval

All owners were previously informed and gave their consent for treatment, sampling, and data recording. All applicable international, national, and/or institutional guidelines for the care and use of animals were followed.

### 2.3. CanL and Other Infectious Diseases Diagnosis

The diagnosis of CanL was assessed by the presence of clinical signs and/or clinicopathological abnormalities and a positive result in the serological test. No direct diagnostic tests were performed to confirm the positivity.

The detection of anti-*Leishmania* antibodies was performed using a commercial enzyme-linked immunosorbent assay—ELISA (Leishmania 96, Agrolabo Spa, Turin, Italy) and the optical density (OD) was measured using a microplate reader (Sirio S, Seac S.r.l., Florence, Italy). The sensitivity and the specificity of the test are reported as 95% and 85%, respectively, while the intra- and inter-variability were 3.5–7.5% and 3.5–8% [26,27]. Following the producer’s instruction, samples were considered positive when above the cut-off of 0.8. Due to the absence of a direct parasitological diagnosis, only dogs with a serological value of more than 1 have been considered for this study. Hematological, biochemical, and urinary parameters were obtained using in-house analyzers (IDEXX point-of-care diagnostics, Westbrook, ME, USA).

For all dogs positive for *Leishmania infantum*, a total clinical score and a total clinicopathological score were calculated by assigning one point to the following signs or alterations: loss of weight, lymph nodes enlargement, skin lesions, spleen enlargement, polyuria and polydipsia, anemia, thrombocytopenia, increased urea and creatinine, increased total protein, decreased albumin, increased gamma globulines, low albumin/globulin (A/G) ratio, and proteinuria.

Positive dogs were also evaluated to rule out the most frequent canine vector-borne infections using the Snap^®^ 4Dx^®^ Plus test (IDEXX Laboratories, Westbrook, ME, USA) to detect *Dirofilaria immitis* antigen, antibodies against *Borrelia burgdorferi*, *Ehrlichia canis/E. ewingii* as well as *Anaplasma phagocytophilum/A. platys*. The test was also applied during the follow-up to all dogs with clinical and or clinicopathological alterations: out of 173 dogs, only 2 tested positive for *Ehrlichia* spp. (3.46%) during the period of follow-up.

### 2.4. Anti-Leishmania Treatment and Relapses

All dogs received 2 mg/kg MIL for 28 days, and 10 mg/kg AL, BID, for a period varying between 2 and 12 months. Ancillary treatments were allowed according to the clinical condition of the dogs.

The retrospective nature of the study did not provide the assessment of the parasitological load using quantitative PCR, useful also to define the relapses. For this reason, in the analysis, the relapsed dogs were considered as those that required more than one MIL-AL treatment.

### 2.5. Follow-Up Period

For all dogs, whatever the number of treatments administered, the follow-up period was calculated from the date of the first treatment to the date of the last visit.

The mean duration of the follow-up period was 5.4 (±1.1) years with a minimum of 3.2 years and a maximum of 9 years. After the first treatment, the follow-up period was calculated as follows: (1) from the date of the first treatment to the date of the last visit for dogs having received only one treatment, (2) from the date of the first treatment to the date of the second treatment for dogs having received more than one treatment. The mean duration of the follow-up period after the first treatment was 8.9 (±10.1) months with a minimum of 3.2 months and a maximum of 71.5 months.

After the second treatment, the follow-up period was calculated: (1) from the date of the second treatment to the date of the last visit for dogs having received only two treatments, (2) from the date of the second treatment to the date of the third treatment for dogs having received more than two treatments. The mean duration of the follow-up period after the second treatment was 38.0 (±20.7) months with a minimum of 7.5 months and a maximum of 96.6 months.

Given the retrospective nature of the work, it was not possible to establish a definite periodicity of controls; so, 12 time points with variable intervals were assessed.

### 2.6. Data Processing and Statistical Analysis

The total clinical and clinicopathological scores obtained in each follow-up were statistically compared with the scores detected at the time of the first diagnosis (T0). Both scores were also compared at each time point to the score at the date of the previous treatment with MIL-AL combination. Improvement of clinical signs or of clinicopathological alterations after each treatment with MIL-AL was defined by the reduction of at least 50% of the total clinical score or of the total clinicopathological score.

A survival analysis (Kaplan–Meier curve) was performed for quantifying the probability of the events occurring during the study follow-up. The following events were considered: (i) decreased and negative ELISA results; (ii) improvement/recovery of the clinical alterations; (iii) improvement/recovery of the clinicopathological alterations; and (iv) relapse of leishmaniasis.

In order to evaluate statistical significance related to the improvement/reduction in clinical and hematobiochemical parameters, a chi-square for trends test was used to compare the data relating to each follow-up.

Statistical analyses were performed using GraphPad InStat Version 3.06 for Windows and MedCalc statistical software Version 7.3.0.1 for Windows.

## 3. Results

### 3.1. Study Population

A total number of 173 dogs of different breeds were included in this study. The dog population was composed of 75 females (43.35%) and 98 males (56.6%). At inclusion, the dogs’ mean age was 3.7 (±2.1) years: in particular, 143 (83%) dogs were under 5 years of age (≤5 years), 28 (16%) were between 5 and 8 years old, and 2 (1%) were over 8 years old.

No dog received previous treatments for CanL, except for three of them treated with a combination of meglumine-antimoniate and allopurinol more than one year prior.

### 3.2. Treatment with MIL-AL Combination

All the 173 dogs included in the study received a first treatment with MIL-AL combination at the inclusion.

During the follow-up, 30 dogs (17.3%) received two treatments, 2 dogs (1.2%) received three treatments and 1 dog (0.6%) received five treatments.

The times of re-administration of MIL-AL during the course of the study are presented in Table 1 and Appendix A.

The mean time interval between the first and the second treatment was 27.2 (±18.3) months with a minimum of 4.8 months and a maximum of 72.8 months. The mean time interval between the second and the third treatment was 16.9 (±11.1) months with a minimum of 9.4 months and a maximum of 29.7 months. Only one dog received a fourth and a fifth treatment in a time interval of 32.7 months.

### 3.3. ELISA

At some point during the follow-up after the first treatment, all examined dogs showed a decreased antibody level (Figure 1). The mean time to achieve a decreased ELISA value after the first treatment was 2.6 (±1.6) months with a minimum of 1.0 and a maximum of 10.5 months.

The survival curve below (Figure 2) shows the probability of survival of dogs whose serological values had not decreased at time t after the first treatment with MIL-AL.

After the first treatment with MIL-AL, the number and percentage of dogs with a decreased ELISA value were calculated for each follow-up (Table 2). Only the data relative to the follow-up of the first treatment were taken into account and if several treatments were performed in the same animal, the last set of data considered were those of the follow-up preceding the next treatment.

All dogs treated at least twice had a decreased serological value at some point during the follow-up after the second treatment. The mean time to observe this decrease was calculated: it was 7.8 (±10.6) months with a minimum of 0.7 months and a maximum of 53.6 months.

After the second treatment with MIL-AL, the number and percentage of dogs with a decreased ELISA value were calculated for each follow-up (Table 3). Only the data relative to the follow-up of the second treatment were taken into account and if several treatments were performed in the same animal, the last set of data considered were those of the follow-up preceding the next one.

Moreover, the dogs that showed a negative ELISA result after the first and the second MIL-AL treatment were evaluated:

After the first treatment, 158 out of the 173 dogs (88.4%) at some point during the follow-up had a negative serological result, and the mean time was 10.5 (±8.3) months with a minimum of 1.4 and a maximum of 57.4 months

After the second treatment, 24 out of the 30 dogs (80%) at some point during the follow-up had a negative serological result and the mean time was 15.6 (±14.4) months with a minimum of 0.8 and a maximum of 53.6 months.

### 3.4. Improvement of Clinical Signs over Time

Most of the dogs at the first diagnosis exhibited clinical signs due to the *L. infantum* infection (Table 4). Each clinical sign after the first treatment with MIL-AL and during the subsequent follow-up was significantly improved (*p* < 0.0001).

After the first treatment with the MIL-Al protocol, 170 out of the 173 dogs (98.3%) exhibited a clinical improvement at some point during follow-up. The mean time of improvement was calculated: 3.0 (±4.9) months with a minimum of 1.0 months and a maximum of 62.9 months. The survival curve below represents the proportion of dogs whose clinical signs had not improved at time t (Figure 3).

Twenty-nine out of the 30 dogs having received a second treatment, at some point during follow-up, showed a reduction in clinical signs. The mean time to improvement after the second treatment was 8.6 (±12.6) months with a minimum of 0.7 and a maximum of 53.6 months. The survival curve below represents the proportion of dogs whose clinical signs had not improved at time t (Figure 4).

### 3.5. Improvement of Clinicopathological Alterations over Time

Most of the dogs at the first diagnosis showed clinicopathological alterations due to the *L. infantum* infection (Table 5). The alterations considered during this analysis, after the first treatment with MIL-AL and during the subsequent follow-up, were significantly improved (*p* < 0.0001).

The dogs with improved clinicopathological alterations at some point during follow-up were identified. They were 171 out of the 173 dogs having received the first treatment. The mean time to observe this decrease was calculated: 4.1 (±10.0) months with a minimum of 1.0 and a maximum of 115.3 months. The survival curve below represents the proportion of dogs whose clinicopathological alterations had not improved at time t (Figure 5).

After the second MIL-AL treatment, 28 out of the 30 dogs showed an improvement in clinicopathological alterations at some point during follow-up. The mean time to observe this event was 11.0 (±15.4) months with a minimum of 0.7 and a maximum of 60.6 months. The survival curve below represents the proportion of dogs whose clinicopathological alterations had not improved at time t (Figure 6).

### 3.6. Relapse of Leishmaniasis

After the first treatment with the MIL-AL protocol, 30 dogs relapsed (17.3%) and 3 dogs had more than one relapse throughout the period of study (Table 6). The mean time without relapse was 90.4 (±2.5) months.

## 4. Discussion

The present study has confirmed that the leishmanicidal activity of Miltefosine in combination with Allopurinol, integrated with case-related supportive therapies, is optimal for the clinical stabilization of dogs affected by CanL, even in a long period of follow-ups.

Above all, the use of the MIL-AL protocol has found excellent compliance by the owners, both for the simple mode of administration and for the limited adverse reactions. In fact, as already demonstrated by several authors [16,28]. Miltefosine has shown good tolerability, with the exception of mild and self-limiting gastrointestinal side effects. None of the 173 dogs examined during the study was excluded due to adverse effects related to therapy, and only 5 subjects showed transient vomit and/or diarrhea.

As already demonstrated, the therapeutic protocol with MIL-AL is not rapid in inducing clinical improvement. A comparative study between the MIL-AL association and meglumine-antimoniate—allopurinol (ME-AL) combination demonstrated that dogs treated with ME-AL achieve clinical improvement faster [21]. However, this is a well-known element linked to the pharmacokinetics of miltefosine, which after its administration, it is characterized by slow absorption and extremely slow elimination [12].

In examined dogs, the antibody value returned within the cut-off after the first therapeutic cycle and, in dogs that did not relapse, they were characterized by a long phase of stabilization. The normalization of clinical and clinicopathological parameters after the first treatment with MIL-AL was simultaneous, with a positive progressive trend over the follow-up period. The only exception regarded the gamma globulins reduction, which took a longer time. The MIL-AL combination was well tolerated and useful in dogs with renal impairment as demonstrated by the improvement of proteinuria and albumin values in most of them.

In this study, 17% of treated dogs required a second MIL-AL treatment in a period varying between 5 months and 6 years. In the same period, however, only three dogs needed further anti-*Leishmania* treatment, a positive element in order to limit the possible pharmacological resistances. The lack of comparative studies assessing the rate of relapses over a long period of follow-up does not allow the determination of whether this incidence represents a favourable outcome. However, it is well known that after the treatment with MIL-AL, the clearance of *Leishmania* in blood, bone marrow, and lymph nodes is not complete, making dogs susceptible to relapses [28].

One of the main findings of the study concerns the faster improvement/healing time of the dogs submitted to one MIL-AL treatment when compared to patients that received more than one therapeutic cycle. This result suggests that the therapeutic efficacy of MIL-AL is less favourable when re-administered after relapse. The authors do not have a clear explanation for this event, especially because the retrospective feature of the study did not allow an in-depth examination of the dogs that relapsed. Indeed, the factors that can determine the lower efficacy of the therapy are several and can be linked to the immune response profile of the dog [29], but especially to the changes of the parasite [23,30,31]. There is no doubt that, in patients requiring more than one treatment, it is useful to alternate the anti-*Leishmania* molecules in order to limit the development of the drug-resistance.

The present study had some limitations. Certainly, the retrospective nature of the analysis has placed some important limits, such as the lack of clinical staging, the reduced uniformity of collateral treatments, the non-uniformity of the follow-up period, and above all, the absence of a control group.

## 5. Conclusions

The study confirmed the effectiveness of MIL-AL used at the standard dosage and its tolerability with only a few self-limiting side effects described. Even if the MIL-AL protocol action was not rapid in dogs that did not relapse, serological and clinical improvement were characterized by a long phase of stabilization and a progressive positive trend. During the period of follow-up, relapses have been reported and the study showed how, in dogs requiring more than one therapeutic cycle, the MIL-AL combination was less favourable in terms of faster time of improvement or healing. In conclusion, the therapeutic protocol with MIL-AL proved to be a good choice in the treatment of dogs affected by CanL when examined for a long period of follow-up.

## Figures and Tables

**Figure 1 pathogens-12-00864-f001:**
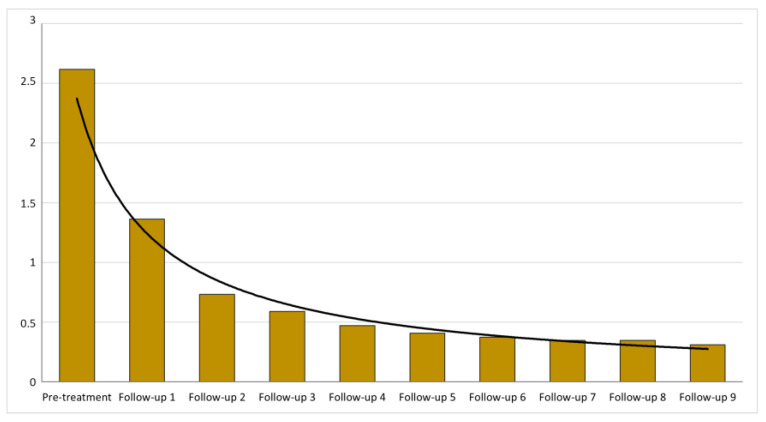
Evolution of average ELISA values during the first nine follow-ups.

**Figure 2 pathogens-12-00864-f002:**
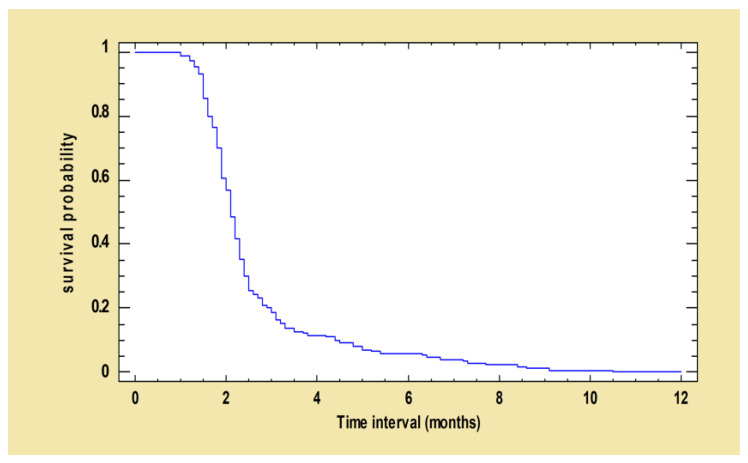
Kaplan–Meier curve: survival function estimated using ELISA decreased after the first treatment. The following survival curve shows the probability of survival of dogs whose ELISA values had not decreased at time t after the first treatment with MIL-AL.

**Figure 3 pathogens-12-00864-f003:**
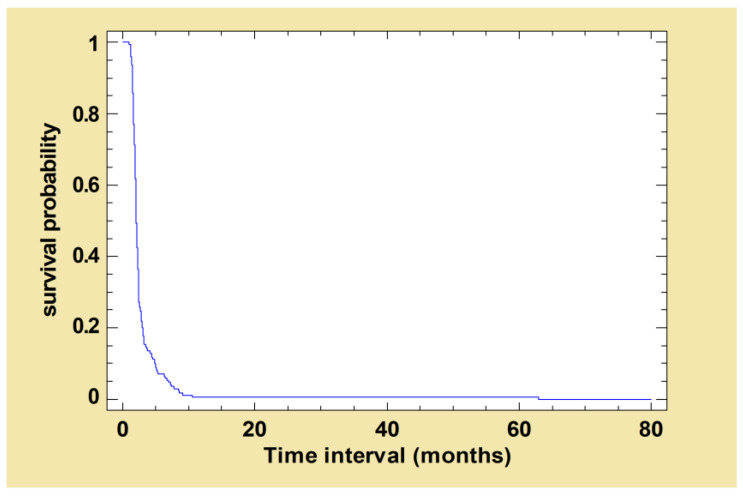
Kaplan–Meier curve: survival function estimated using total clinical score decreased after the first treatment. The following survival curve shows the probability of survival of dogs whose total clinical score had not decreased at time t after the first treatment with MIL-AL.

**Figure 4 pathogens-12-00864-f004:**
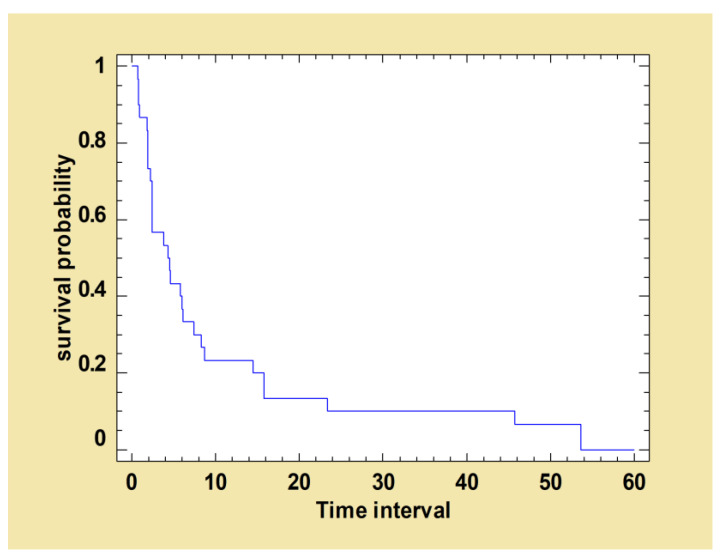
Kaplan–Meier curve: survival function estimated using total clinical score decreased after the second treatment. The following survival curve shows the probability of survival of dogs whose total clinical score had not decreased at time t after the second treatment with MIL-AL.

**Figure 5 pathogens-12-00864-f005:**
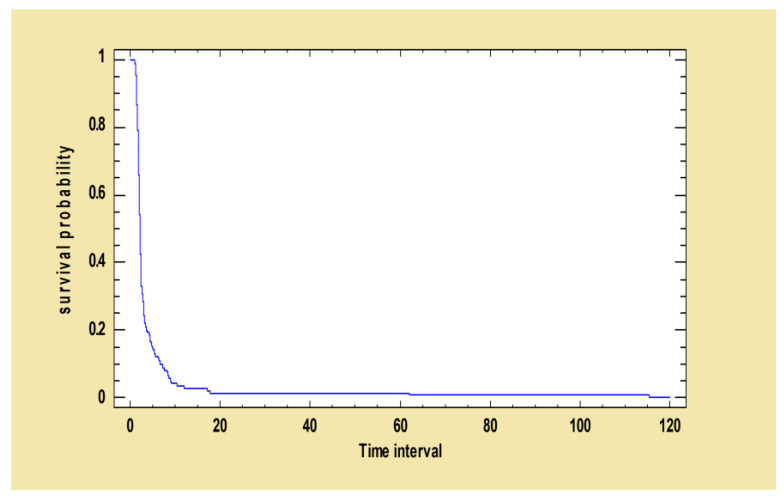
Kaplan–Meier curve: survival function estimated using total clinicopathological score decreased after the first treatment. The following survival curve shows the probability of survival of dogs whose total clinicopathological score had not decreased at time t after the first treatment with MIL-AL.

**Figure 6 pathogens-12-00864-f006:**
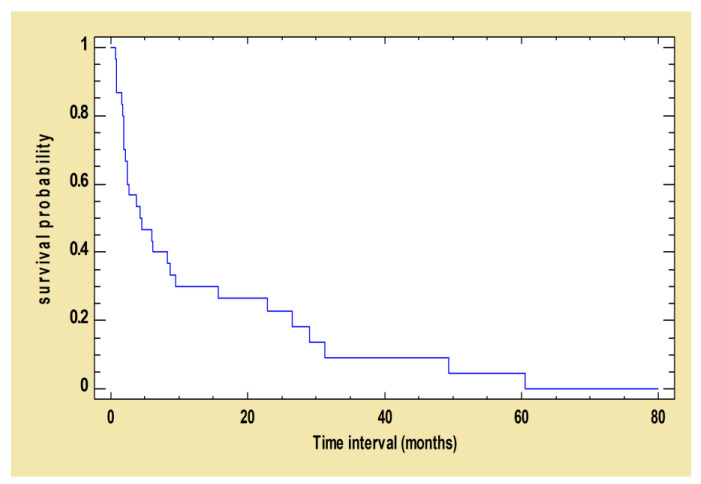
Kaplan–Meier curve: survival function estimated using total clinicopathological score decreased after the second treatment. The following survival curve shows the probability of survival of dogs whose total clinicopathological score had not decreased at time t after the second treatment with MIL-AL.

**Table 1 pathogens-12-00864-t001:** Number of dogs requiring re-administration of MIL-AL therapy during each time point assessed.

Time Point	Re-Administration of MIL-AL N. of Dogs (%)
Follow-up 1	1 (0.6%)
Follow-up 2	2 (1.2%)
Follow-up 3	9 * (5.2%)
Follow-up 4	8 * (4.6%)
Follow-up 5	5 * (2.9%)
Follow-up 6	2 (1.2%)
Follow-up 7	4 * (2.4%)
Follow-up 8	1 * (0.6%)
Follow-up 9	2 * (1.2%)
Follow-up 12	1 (0.6%)

* Three dogs received more than one re-administration of MIL-AL: one dog on follow-ups 3 and 4, one dog on follow-ups 4 and 9, 1 dog on follow-ups 3, 5, 7, and 9.

**Table 2 pathogens-12-00864-t002:** ELISA trend after the first MIL-AL treatment.

Follow-Up	Dogs (n.)	Dogs with Decreased ELISA Value (n. and %)	Dogs with Increased ELISA Value (n. and %)
1	161	161 (100%)	0 (0%)
2	169	169 (100%)	0 (0%)
3	162	161 (99.4%)	1 (0.6%)
4	155	154 (99.4%)	1 (0.6%)
5	149	149 (100%)	0 (0%)
6	142	140 (98.6%)	2 (1.4%)
7	117	115 (98.3%)	2 (1.7%)
8	91	90 (98.9%)	1 (1.1%)
9	54	54 (100%)	0 (0%)
10	29	29 (100%)	0 (0%)
11	14	14 (100%)	0 (0%)
12	6	6 (100%)	0 (0%)
13	1	1 (100%)	0 (0%)
14 *	1	1 (100%)	0 (0%)

* At most, 14 follow-up visits were performed after the first treatment.

**Table 3 pathogens-12-00864-t003:** ELISA trend after the second MIL-AL treatment.

Follow-Up	Dogs (n.)	Dogs with Decreased ELISA Value (n. and %)	Dogs with Increased ELISA Value (n. and %)
1	30	28 (93.3%)	2 (6.7%)
2	27	27 (100%)	0 (0%)
3	25	24 (96.0%)	1 (4.0%)
4	20	20 (100%)	0 (0%)
5	16	16 (100%)	0 (0%)
6	7	7 (100%)	0 (0%)
7	6	6 (100%)	0 (0%)
8	5	5 (100%)	0 (0%)
9	5	5 (100%)	0 (0%)
10	2	2 (100%)	0 (0%)
11	2	2 (100%)	0 (0%)
12 *	1	1 (100%)	0 (0%)

* At most, 12 follow-up visits were performed after the second treatment.

**Table 4 pathogens-12-00864-t004:** Prevalence of clinical signs before and after the first treatment with MIL-AL.

Clinical Signs	Pre-Treatment		First Follow-Up	
	N. of Dogs	Prevalence % (95% CI) *	N. of Dogs	Prevalence % (95% CI) *
Loss of weight	153	88.43 (83.67–93.20)	19	10.98 (6.52–15.64)
Lymph nodes enlargement	139	80.35 (74.43–86.27)	104	60.12 (52.82–67.41)
Skin lesions	96	55.49 (48.09–62.9)	5	2.89 (0.39–5.39)
Spleen enlargement	21	12.14 (7.27–17.01)	23	13.29 (8.24–18.35)
Polyuria and polydipsia	44	25.43 (18.94–31.92)	2	1.16 (0.00–2.75)

* The analysis was performed for the first 9 follow-ups.

**Table 5 pathogens-12-00864-t005:** Prevalence of clinicopathological alterations before and after the first treatment with MIL-AL.

Clinicopathological Alterations	Pre-Treatment		First Follow-Up	
	N. of Dogs	Prevalence % (95% CI) *	N. of Dogs	Prevalence % (95% CI) *
Anemia	155	89.60 (85.05–94.15)	12	6.94 (3.15–10.72)
Thrombocytopenia	64	36.99 (29.80–44.19)	7	4.05 (1.11–6.98)
Increased urea	86	49.71 (42.26–57.16)	7	4.05 (1.11–6.98)
Increased creatinine	105	60.69 (53.42–67.97)	9	5.20 (1.89–8.51)
Increased total protein	73	42.20 (34.84–49.56)	17	9.83 (5.39–14.26)
Decreased albumin	105	60.69 (53.42–67.97)	33	19.08 (13.22–24.93)
Increased gamma globulines	132	76.30 (69.96–82.64)	83	47.98 (40.53–55.42)
Low A/G ratio	34	19.65 (13.73–25.57)	12	6.94 (3.15–10.72)
Proteinuria	83	47.98 (40.53–55.42)	31	17.92 (12.2–23.63)

* The analysis was performed for the first 9 follow-ups.

**Table 6 pathogens-12-00864-t006:** Number of relapses and mean time to relapse.

Number of Relapses	Dogs with Available Data	Dogs with Relapses (n. and %)	Mean Time to Relapse (±months) *
1st	173	30 (17.3%)	27.2 (±18.3)
2nd	30	3 (10.0%)	16.9 (±11.1)
3rd	3	1 (33.3%)	8.7 **
4th	1	1 **	32.7 **

* The time to relapse was calculated between the two last administrations of MIL-AL. ** Value for 1 dog.

## Data Availability

All data are stored by the authors and are always available.

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
