# Peer review of "Retrospective Long-Term Evaluation of Miltefosine-Allopurinol Treatment in Canine Leishmaniosis"

_pathogens, 2023, doi:10.3390/pathogens12070864_

Round 1

Reviewer 1 Report

Introduction is well written and appropriate references have been cited.

Regarding study population, Authors should describe better dog population used as serum source: it is unclear whether sera from true positive and true negative dogs to Leishmania infantum infection were used for the study; at this regard, did Authors use direct diagnostic test as molecular analysis, parasitological analysis conducted on bone marrow cells, and/or the gold standard indirect test IFAT in order to assess the antibody titer?

Authors wrote “The detection of anti-Leishmania antibodies was performed by commercial enzyme-linked immunosorbent assay – ELISA (Leishmania 96, Agrolabo Spa, Turin, Italy) and the optical density (OD) was measured using a microplate reader (Sirio S, Seac S.r.l., Florence, Italy).” What about the species-specificity of the test? What about the intra- and inter-varianility test?

Regarding statistical analysis, did Authors check the normal distribution of data by a Normality test?

In Tables 2 and 3, Authors wrote “ELISA trend after the first / the second MIL-AL treatment”, and they reported “decreased titer” in the columns; however, the ELISA results showed a S/P percentage that did not reflect the real antibody titer.

In method section Authors wrote “Positive dogs were also evaluated to rule out the most frequent canine vector borne infections using the Snap® 4Dx® Plus test (IDEXX Laboratories, Westbrook, USA) to de tect Dirofilaria immitis antigen, antibodies against Borrelia burgdorferi, Ehrlichia canis/E. ewingii as well as Anaplasma phagocytophilum/A.platys. The test was also applied during the follow up to all dogs with clinical and or clinicopathological alterations”

Please report the results of these tests.

Discussion is clear and Authors well justify and discuss obtained results.

I suggest to add a conclusion section in which Authors summarize the rationale and the main findings of the study as well as to emphasize the significance of the study.

Author Response

Dear Reviewer,

thank you for all comments and suggestions.

Please see the attachment for our reply.

Regards

Reviewer 2 Report

The work "Retrospective long-term evaluation of Miltefosine/Allopurinol 1 treatment in canine leishmaniosis" is interesting, addressing the effect of miltefosine with allopurinol therapy in dogs with canine leishmaniasis. However, I believe that it needs to have its results better explained, as suggested below:

The language needs to be revised.

Replace "leishmaniosis" with "leishmaniasis".

Abstract:

What is CanL? To specify.

Introduction:

Join the first two paragraphs.

Join third and fourth paragraphs.

Join fifth, sixth and seventh paragraphs.

Materials and Methods:

Line 87: The abbreviation CanL has already been shown, therefore, it must be used throughout the text.

Lines 93 - 96: Ethics committee acceptance number.

Line 110: What is A/G ratio inversion? To specify.

Lines 125 – 145: Suggestion:

For all dogs, whatever the number of treatments administered, the follow-up period was calculated from the date of the first treatment until the date of the last visit. The mean duration of follow-up period was 5.4 (± 1.1) years with a minimum of 3.2 128 years and a maximum of 9 years. After the first treatment, the follow-up period was cal-129 culated as following: 1) from the date of the first treatment until the date of the last visit for dogs having received only one treatment, 2) from the date of the first treatment until the date of the second treatment for dogs having received more than one treatment. The mean duration of follow-up period after the first treatment was 8.9 (± 10.1) months with a minimum of 3.2 months and a maximum of 71.5 months.

After the second treatment the follow-up period was calculated: 1) from the date of the second treatment until the date of the last visit for dogs having received only two treatments, 2) from the date of the second treatment until the date of the third treatment for dogs having received more than two treatments. The mean duration of follow-up period after the second treatment was 38.0 (± 20.7) 142 months with a minimum of 7.5 months and a maximum of 96.6 months. 143

Given the retrospective nature of the work, it was not possible to establish a definite 144 periodicity of controls, so 12 time points with variable intervals were assessed.

Lines 146 – 153: Join paragraphs.

Results:

Line 176 and Lines 186 - 187: Discordant information from the abstract.

Tabla 1: Difficult to understand. Time of re-administration of MIL-AL must be presented in days or months. And in my opinion it should be placed individually by animal.

What is “Follow-up”?

Lines 189 – 192: It is necessary to show the reduction in antibody levels that has been cited. Through a figure.

Figure 1: What is this curve? Hard to understand. It needs to be explained much better.

Lines 225 – 226: I suggest adding one more column in Table 4 showing the prevalence of each clinical signs after the first treatment, so that we can compare with the pre-treatment data. It would also be interesting to add more columns to Table 4, also showing data after the second and third treatments.

Line 232: 62.9 months? Is it correct?

Figures 2, 3, 4 and 5: Difficult to understand. To explain better.

Lines 246 – 249: I suggest adding one more column in Table 5 showing the prevalence of each clinicopathological signs after the first treatment, so that we can compare with the pre-treatment data. It would also be interesting to add more columns to Table 5, also showing data after the second and third treatments.

Lines 271 – 275: Present the data in a table.

Author Response

(The authors gave the same response as above.)
